# MHKD: Multi-step Hybrid Knowledge Distillation for Low-resolution Whole Slide Images Glomerulus Detection

Xiangsen Zhang
*NWPU*
Xi'an, China
xszhang116@gmail.com

Longfei Han
*BTBU*
Beijing, China
longfeihan@btbu.edu.cn

Chenchu Xu
*Anhui University*
Hefei, China
xcc@ahu.edu.cn

Zhaohui Zheng
*Xijing Hospital*
Xi'an, China
zhengzh@fmmu.edu.cn

Jin Ding
*Xijing Hospital*
Xi'an, China
dingjin@fmmu.edu.cn

Xianghui Fu
*Xijing Hospital*
Xi'an, China
fuxianghui0225@126.com

Dingwen Zhang
*NWPU, Xijing Hospital*
Xi'an, China
zhangdingwen2006yyy@gmail.com

Junwei Han
*NWPU*
Xi'an, China
jhan@nwpu.edu.cn

*Abstract*—Glomerulus detection is a critical component of renal histopathology assessment, essential for diagnosing glomerulonephritis. To mitigate the increasing workload on pathologists, AI-assisted diagnostic methods based on high-resolution digital pathology whole slide images have been developed. However, these current AI-assisted approaches are limited to high-resolution whole slide images, necessitating expensive digital scanner equipment, high image storage costs, and significant computational complexity. To address this limitation, this paper pioneers a method for facilitating glomerulus detection in low-resolution human kidney pathology images. Specifically, we propose a novel multi-step hybrid knowledge distillation method. Our method distills both the global features and the semantic information through a hybrid knowledge distillation strategy that integrates offline and online knowledge distillation, where the information from high-resolution pathological images is successively transferred to student model from the global features in the shallow network layers to the semantic information of the back-end through a multi-step training strategy. Experimental results on two datasets show that the proposed method achieves effective detection outcomes for low-resolution kidney pathology images. Compared to other state-of-the-art detection techniques, our method achieves an $AP_{0.5:0.95}$ improvement of **23.1%** on the private LN dataset and **15.9%** on the public HUBMAP dataset.

*Index Terms*—glomerulus detection, hybrid knowledge distillation, low-resolution pathology image, multi-step training strategy

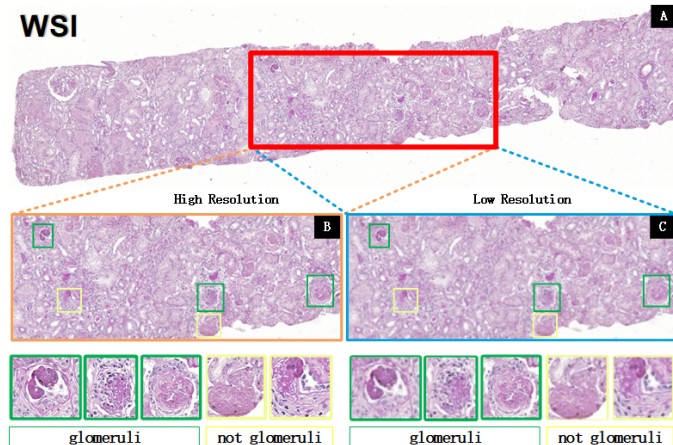

Fig. 1. Examples of different resolutions in the same area. The green box represents the glomerular region and the yellow box is the non-glomerular region that appears similar to a glomerulus. In low-resolution pathological images, the boundary of glomeruli in the green box area is not obvious, and the yellow box area is easily misjudged as a glomerulus.

## I. Introduction

Glomerulonephritis is a common and potentially life-threatening kidney disease primarily affecting the structure and function of the glomeruli. Pathological diagnosis is the gold standard for the diagnosis of glomerulonephritis, enabling the detection of diseased glomeruli and the quantitative assessment of nephritis severity. However, this diagnostic process necessitates highly skilled pathologists and is labor-intensive, failing to meet the current demands. In recent years, AI-assisted diagnosis has been integrated into the detection of pathological images, providing a potential solution [5], [11], [12] to reduce the burden on pathologists who must continuously analyze large batches of images. Various methods for glomerulus detection [2], [3], [4], [30], [31] have demonstrated commendable diagnostic performance.

However, the current methods have limited application scenarios. These approaches are developed based on high-resolution pathological images, necessitating costly scanners and demanding scanning accuracy. Consequently, they are only suitable for relatively advanced regions rather than less developed areas and countries. According to the survey, the price

This work is supported by the National Natural Science Foundation of China (No. 62202015, 2272468), Key-Area Research and Development Program of Shaanxi Province under Grant 2023-ZDISF-41, Anhui Provincial Key R&D Programmes (2023s07020001).eCF Paper Id: 773204 has successfully completed the IEEE Electronic Publication Agreement.

of automatic digital scanner equipment used to acquire high-resolution pathology images is generally 20 to 40 times higher than the price of equipment used to acquire low-resolution pathology images, or even more. Additionally, the time and storage costs associated with obtaining high-resolution pathological images are significant. Under these circumstances, there is a great demand for pathological diagnosis based on low-resolution pathological images.

Unfortunately, there are great challenges in performing glomerulus detection based on low-resolution digital images. Sensitivity to subtle structural changes and color differences in tissue is a key indicator of a pathologist's expertise and is crucial for disease diagnosis. However, such cues are often lost in low-resolution pathological images. As illustrated in Fig.1, the comparison between high-resolution and low-resolution kidney pathology images reveals that glomeruli, marked by green boxes, have blurred boundaries in low-resolution images, making them easily overlooked. Additionally, non-glomerular tissues, marked by yellow boxes, are often confused with glomeruli in low-resolution images due to the lack of discernible details. The nuclei of mesangial cells and the thickness of capillaries are critical indicators for assessing abnormalities, yet this information is difficult to discern in low-resolution images. Due to the aforementioned issues, applying low-resolution whole slide images (WSIs) directly to current detection methods [7], [8], [9], [10] significantly degrades detection performance compared to high-resolution WSIs.

To address this problem, this paper proposes a novel multi-step hybrid knowledge distillation (MHKD) method. The core concept is to employ knowledge distillation to enhance the student model's capability to extract features from low-resolution pathological images by transferring information from high-resolution pathological images in the teacher model to the student model. This knowledge distillation process compensates for the missing details in low-resolution pathological images, resulting in detection performance that closely approximates that of high-resolution images. However, due to the specific nature of pathological input images, conventional knowledge distillation methods do not achieve optimal results. Firstly, the high-resolution pathological image information in the teacher model is crucial for the distillation task, so we need to ensure the extraction and transfer of high-resolution information at the forward end, and on this basis to achieve efficient information transmission at other locations. In addition, different from natural images, the task-related features and discriminative small details of kidney pathological images are densely concentrated in small local areas, and there is a large information gap between pathological images of different resolutions. It is difficult to ensure the effectiveness of critical knowledge transfer by distillation of multiple positions of the model at the same time.

Firstly, hybrid knowledge distillation integrates offline distillation and online distillation to optimize the distillation performance. Typically, online distillation surpasses offline distillation, but direct application in our specific task is not suitable. We discovered that training solely with online distillation would compromise the feature extraction capability of the teacher model due to difference in input pathological image resolutions. To address this issue, we exclusively employ offline distillation in the backbone part of the teacher model.

Secondly, a multi-step training strategy is devised for the distillation task involving varying resolutions of input pathological images. We divide distillation locations into two main groups: global features at the shallow network layers and semantic information at the back-end. Semantic information includes local features and output. The Basic Feature Adaptive module has been designed to enhance the adaptability of basic global features, enabling them to assimilate knowledge effectively from the teacher network. To sequentially transfer information from high-resolution pathological images through basic features, adaptive features, local features, and semantic information to the student model, we adopt a multi-step training approach.

Overall, our contributions are summarized as follows:

- To the best of our knowledge, we are the first to propose the task of glomerulus detection on low-resolution human kidney pathology images and propose a novel multi-step hybrid knowledge distillation method for this task.
- To effectively optimize knowledge transfer across different resolution inputs, we propose a hybrid knowledge distillation strategy that integrates both online and offline approaches. The offline distillation of forward-end global features ensures the feature extraction ability of the teacher model, and the online distillation of backward-end semantic information realizes the real-time guidance of the teacher model and shortens the distance between the student model and the teacher model.
- To effectively tackle the issue of insufficient discriminative information in low-resolution pathological images, we present a multi-step training strategy. This strategy sequentially guides distillation from global features to semantic information, facilitating the systematic transfer of knowledge from high-resolution pathological images to the student model.
- For the task of glomerulus detection on low-resolution digital pathology images of human kidneys, our proposed method demonstrates superior performance on both a private LN dataset and the public HUBMAP dataset compared to the state-of-the-art detection methods.

## II. RELATED WORKS

### A. Diagnosis of Kidney Disease by Medical Imaging

In recent years, the advancement of computer technology has led to an increasing number of studies applying computer technology for the analysis and processing of digital kidney images. In the diagnosis of renal disease digital images, according to the source of the image, the diagnostic methods can be divided into US-based methods, such as [26], CT-based methods, such as [18], MRI-based methods, such as [27], and WSI-based methods, such as [4]. The WSI-based approach is

the one to watch due to renal pathology being considered the gold standard for diagnosing renal diseases. Evidence suggests that patients who undergo renal biopsy have higher kidney survival rates compared to those who do not [1].

There are two ways for glomerulus detection of a single WSI in the WSI-based approach: one is the slider method. Gal lego et al. [3] proposed a method that slices WSIs into patches using a fixed-size sliding window approach followed by patch classification into glomeruli or background using CNNs. The classifiers in it can be changed according to the task. The second is localization before classification. Zheng et al. [4] divided their approach for detecting glomeruli into two stages: initial localization on low-resolution WSIs followed by interception of high-resolution local areas based on this localization for subsequent classification. In addition, multimodal combination of the above methods is also helpful for glomerulus detection, such as WSIs with multiple stains. Yoshimasa Kawazoe et al. [2] employed different dyes for WSI staining and used a sliding window technique for WSI detection.

However, these methods all require high-resolution WSIs. Therefore, we propose a novel task of glomerulus detection based on low-resolution human kidney digital pathology WSIs to reduce the dependence on medical resources.

### B. Knowledge Distillation

Knowledge distillation, a method often used for model compression, is an effective technique for transferring knowledge from large-scale teacher models to small-scale student models. At present, most knowledge distillation is applied to classification tasks [21], [22]. It was first proposed by Hinton et al. [21], which uses the output as a soft label to transfer the dark knowledge from a large teacher network to a small student network for classification. Recently, several works have successfully applied knowledge distillation to object detection tasks. Chen et al. [23] was the first to apply knowledge distillation to object detection through the combination of feature and prediction. Subsequently, region-specific feature distillation has been proposed, such as Li et al. [24] proposed a scheme to imitate features within regions in Faster R-CNN. In addition, with the proposal of Deep Mutual Learning (DML) [25] and [29], distillation can be divided into offline distillation and online distillation according to whether the teacher model updates parameters during the training process. The main advantage of the offline distillation [14], [15] is its simplicity and ease of implementation. however, the disadvantage is that complex high-capacity teacher models and huge training time cannot be avoided. In online distillation [16], [17], both the teacher model and the student model are updated simultaneously. However, existing online approaches usually fail to address the problem of high-ability teachers in online Settings. In this paper, we adopt a distillation strategy combining the two.

## III. PROPOSED METHOD

### A. Preliminaries

The object detection algorithm for pathological images employs a two-stage approach, where proposals are extracted from the feature maps to acquire regions of interest (ROIs), followed by classification and regression on each ROI. Consequently, the overall loss function for object detection can be formulated as follows.

$$\mathcal{L}_{\text{detect}} = \mathcal{L}_{\text{cls}} + \mathcal{L}_{\text{reg}}, \tag{1}$$

where $\mathcal{L}_{\text{cls}}$ is all classification loss, $\mathcal{L}_{\text{reg}}$ is all regression loss. Considering the strategy of knowledge distillation, the range of $\mathcal{L}_{\text{detect}}$ needs to be further determined. If the strategy is offline distillation, $\mathcal{L}_{\text{detect}}$ only contains the loss of student model. If the strategy is online distillation, $\mathcal{L}_{\text{detect}}$ contains both the loss of student model and teacher model.

To the best of our knowledge, no prior method has attempted knowledge distillation for object detection in pathological images with varying resolutions. Existing approaches to knowledge distillation in object detection tasks typically focus on selecting information from two locations: features and outputs. Hence, the formulation of the knowledge distillation loss function can be expressed as follows.

$$\mathcal{L}_{\text{KD}} = \mathcal{L}_{\text{fea}} + \mathcal{L}_{\text{out}}, \tag{2}$$

where $\mathcal{L}_{\text{fea}}$ is the feature distillation loss of different regions or layers and $\mathcal{L}_{\text{out}}$ is the output distillation loss. The feature distillation can be further categorized based on the location of features. Additionally, $\mathcal{L}_{\text{out}}$, also known as the soft loss, is the loss computed between the output of the teacher model and the output of the student model.

The overall training loss comprises the detection loss $\mathcal{L}_{\text{detect}}$ and the knowledge distillation loss $\mathcal{L}_{\text{KD}}$, thus can be formulated as follows:

$$\mathcal{L}_{\text{all}} = \mathcal{L}_{\text{detect}} + \lambda_{KD} \cdot \mathcal{L}_{\text{KD}}, \tag{3}$$

where $\lambda_{KD}$ is a weight coefficient for knowledge distillation. In our method, we select information for distillation in two locations: global feature (basic feature and adaptive feature), semantic information (local feature and output).

### B. Hybrid Knowledge Distillation

The proposed hybrid knowledge distillation method integrates both offline and online distillation strategies. With different resolutions of the input pathological images, hybrid knowledge distillation not only can maintain that the feature extractor in the input end of the teacher model can efficiently extract the features of high-resolution pathological images, but also ensure that the semantic information in the output end gets real-time guidance from the teacher model, so as to shorten the gap between the student model and the teacher model. As show in Fig. 2, our proposed method is divided into two parts: Global Feature Offline Distillation for the first half and Semantic Information Online Distillation for the second half.

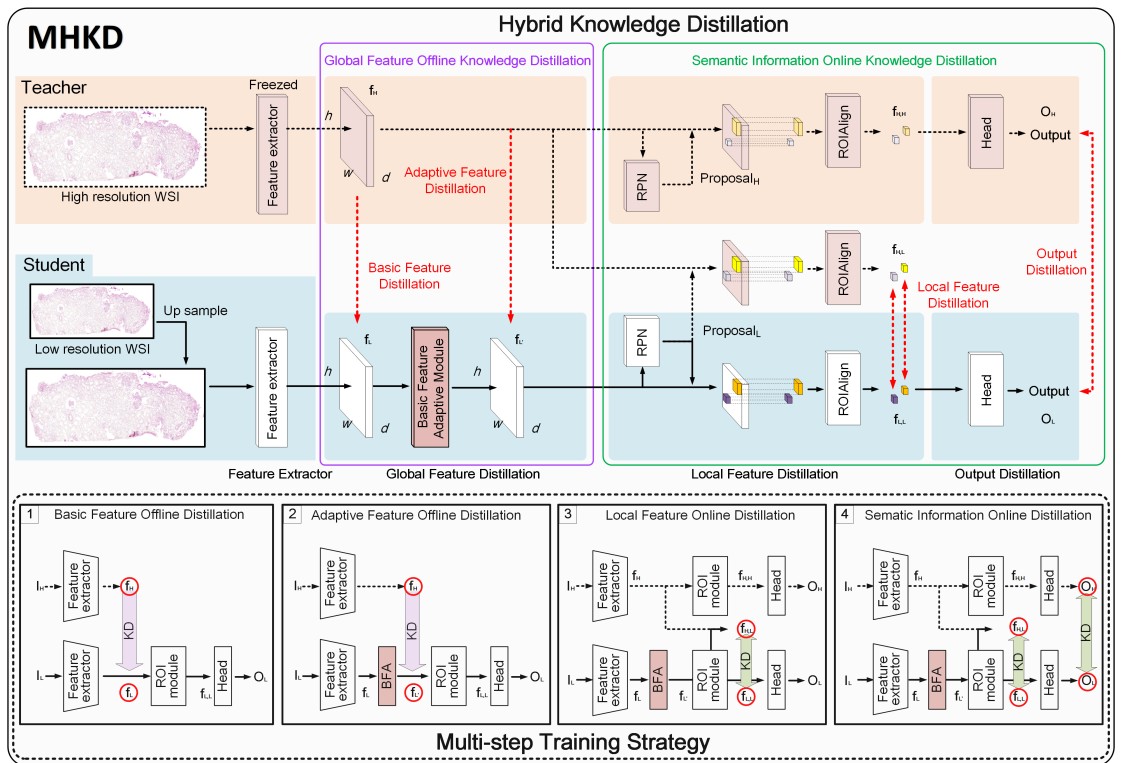

Fig. 2. Illustration of our proposed method MHKD. The top shows the framework of MHKD and the Hybrid Knowledge Distillation strategy, and the bottom shows the Multi-step Training Strategy. The orange path represents the teacher model and the blue path represents the student model.

Before inputting images into the network, low-resolution pathological images undergo upsampling to match the high-resolution counterparts. This procedure guarantees a consistent receptive field for the global features extracted by both the teacher and student models.

*1) Global Feature Offline Distillation:* The role of global feature offline distillation module is to transfer the high-resolution pathological image information from the teacher model to the student model through distillation at the global feature level. This process aims to enhance the representation ability of the global features extracted by the student model and supplement the discriminative information.

As depicted in Fig. 2, our method employs offline distillation on the global feature within the global feature offline distillation module. The global feature represents comprehensive information extracted by the feature extractor from the input image, serveing as a fundamental component for all tasks. Hence, we perform offline knowledge distillation to ensure that the feature extraction capability of the already trained teacher model remains unaffected by the student model. By mimicking the teacher model's global features $\mathbf{f_H}$, our approach enables the student model to obtain the basic feature $\mathbf{f_L}$that is closer to the teacher model. Both the basic feature and the adaptive feature are global features of student model, and the adaptive feature will be explained in the next section. Although online distillation typically yields better results with identical inputs, it is unsuitable for our task at this time due to low-resolution

student inputs that would degrade teacher performance. Consequently, we propose hybrid knowledge distillation which incorporates offline distillation specifically for global feature distillation. To elaborate further, we feed the teacher input $\mathbf{I_H}$ into the frozen backbone of the teacher model to extract high-quality global features $\mathbf{f_H} \in \mathbb{R}^{w \times h \times d}$. Simultaneously, we pass an up-sampled version of student input $\mathbf{I_L}$ through the student model's backbone to obtain corresponding global features $\mathbf{f_L} \in \mathbb{R}^{w \times h \times d}$. To guide pixel-level mimicry between student's global feature $\mathbf{f_L}$ and its approximation towards $\mathbf{f_H}$, we employ the mean square error (MSE) loss as our chosen distillation loss function.

$$\mathcal{L}_{\text{basic}} = \frac{1}{N} \sum_{n=1}^{N} (\mathbf{f_{H,n}} - \mathbf{f_{L,n}}), N = w \times h \times d \quad (4)$$

*2) Semantic Information Online Distillation:* In the semantic information online distillation phase, we optimize the local features based on the proposals provided by RPN and optimize the final output. The local features, referred to as ROI features, represent the suspected target regions. Specifically, both teacher and student models predict proposals: $Proposal_H$ and $Proposal_L$, respectively. The teacher model generates a series of ROI features $\mathbf{f_{H,H}}$ using $\mathbf{f_H}$ and $Proposal_H$, which are then used to predict $\mathbf{O_T}$. Similarly, the student model generates a series of ROI features $\mathbf{f_{L,L}}$ using $\mathbf{f_L}$ and $Proposal_L$, which are utilized for predicting $\mathbf{O_S}$. To ensure that the local features of the student model imitate the target features within

the same regions, we employ identical coordinates for region selection. The teacher model generates a series of $\mathbf{f_{H,L}}$ based on $\mathbf{f_H}$ and $Proposal_L$, which represent the proposals from the student model. Subsequently, pixel-wise MSE loss is utilized to compute the losses between $\mathbf{f_{H,L}}$ and $\mathbf{f_{L,L}}$.

$$\mathcal{L}_{\text{local}} = \frac{1}{M} \sum_{m=1}^{M} (\mathbf{f}_{H,L}^m - \mathbf{f}_{L,L}^m) \tag{5}$$

where M is the number of pixels in each ROI feature.

In addition to local feature distillation, we also introduce distillation to guide the student model to produce more accurate predictions at the final prediction output. In this method, the teacher's output $\mathbf{O_T}$ is transformed into the same data format as the ground truth (GT), replacing the role of GT in normal training. This enables the student model to learn from the teacher's predictions directly. In the semantic online distillation part, we introduce two distillations, one for local features and another for the output. These locations contain valuable semantic information that is closely related to the final prediction. By adopting the online distillation strategy close to the output, the teacher model can better explore the relationship between teachers and students, realize real-time guidance, effectively reduce the gap between the teacher and student models, and improve the performance of the student model. Consequently, our proposed hybrid knowledge distillation method seamlessly integrates offline and online approaches while incorporating global feature extraction, local feature extraction, and output-based distillations.

### C. Multi-Step Training Strategy

To improve the distillation effect, we divide the training into four steps: basic feature offline distillation, adaptive feature offline distillation, local feature online distillation, and sematic information online distillation. The reason is that progressive distillation training can effectively ensure that the high-resolution pathological image information in the teacher model can be gradually transferred to the student model.

*1) Concrete Steps:* The hybrid knowledge distillation method we mentioned earlier divides the whole framework into two parts. In the global feature offline distillation part, the performance of the teacher model is not degraded. The semantic online distillation part can effectively reduce the gap between the teacher model and the student model.

However, in the global feature offline distillation part, there is still a certain gap between global feature $\mathbf{f_L}$ and $\mathbf{f_H}$ after direct distillation due to significant information gaps. Therefore, we introduce adaptive feature offline distillation to divide the global feature offline distillation into two steps: basic feature offline distillation and adaptive feature offline distillation. Specifically, we distilled the basic feature $\mathbf{f_L}$ and then performed adaptive transformation to obtain $\mathbf{f_L'}$, and then distilled it again. Consequently, the global features of the student model are adaptively enhanced and the information transfer from the teacher model is obtained twice respectively. The global features obtained by the student model exhibit a higher degree of proximity to those acquired by the teacher model.

Specifically, in basic feature offline distillation, the teacher model is frozen and the student model is trained, and the loss includes the detection loss and the global feature distillation loss. In adaptive feature offline distillation, the teacher model is frozen and the student model is trained, and the loss includes detection loss and global feature distillation loss. A new module, Basic Feature Adaptive, is introduced to generate adaptive global features in student model. In semantic feature online distillation, the feature extractor of the teacher model is frozen, the rest of the teacher model is trained, and the student model is trained. the loss includes detection loss and local feature distillation loss. In semantic information online distillation, the feature extractor of the teacher model is frozen, the rest of the teacher model is trained, and the student model is trained. And the loss includes detection loss, local feature distillation loss, and output loss.

*2) Basic Feature Adaptive Module:* To reduce the gap between the global features of the student model and the global features of the teacher model, we introduce the Basic Feature Adaptive (BFA) module after the extracted global features $f_L$ to obtain better global features. Considering that the discrepancy between the teacher and student models lies in their input image resolutions, our BFA module is modified based on SRResnet [28], with adjustments made to parameters such as channel numbers and scaling factors.

The SRResNet model is a deep learning architecture designed for image super-resolution tasks. It utilizes residual connections and convolutional neural networks (CNN) to achieve high-quality upscaling of low-resolution images, enhancing image details and clarity. By employing sub-pixel convolution, the model effectively addresses the challenge of increasing image size without compromising pixel density. Therefore, as a feature adaptive module based on this modification, we can better enrich the global feature of low-resolution images.

## IV. EXPERIMENT RESULTS

### A. Dataset

To facilitate the exploration of the role of knowledge distillation for detection tasks on low-resolution pathological images, the dataset is downsampled so that WSIs can be directly used as input to the network in this paper.

*1) LN Dataset:* The LN dataset contains 349 kidney biopsy WSIs from 163 patients, collected between 2011 and 2019, with varied ages, genders, and degrees of lupus nephritis. This dataset is focused on lupus nephritis and can be utilized for glomerulus detection and the diagnosis of lupus nephritis. Glomeruli were classified into five categories: 'Slight', 'Severe', 'Fibrosis', 'Incomplete', and 'Uncertain'.

The data were obtained from the Chinese Systemic Lupus Erythematosus (SLE) Cohort database of Xijing Hospital and were approved by the Institutional Review Board of the Ethics Committee of Xijing Hospital (KY20223382-1) for this study.

*2) HuBMAP Dataset:* The HuBMAP dataset [32] is a public dataset. The HuBMAP data used in this hackathon includes 11 fresh frozen and 9 Formalin Fixed Paraffin Embedded (FFPE) PAS kidney images. Glomeruli segmentation annotations exist for 15 tissue samples. Each sample is large in size and includes a substantial number of glomeruli. In this paper we divided it into 171 pathological digital imagesd for one-class glomerular detection only.

*B. Setting*

We utilized a Linux server equipped with a single NVIDIA GeForce RTX 3090 to construct the trained model for glomerulus detection. The high-resolution branch, serving as the teacher model, was pre-trained and its backbone was frozen. Conversely, the low-resolution branch, acting as the student model, was not pre-trained. In order to facilitate the exploration of ways to improve the input of low-resolution images, we down sampled the input images. For LN dataset, we set the input size of the high-resolution branch to $768 \times 2304$ pixels while maintaining a resolution of $256 \times 768$ pixels for the low-resolution branch. For HUBMAP dataset, we set the input size of the high-resolution branch to $1024 \times 1024$ pixels while maintaining a resolution of $256 \times 256$ pixels for the low-resolution branch. The initial learning rate was set at $1 \times 10^{-3}$, and every five epochs witnessed a decrease in learning rate by a factor of 0.1. The batch size is assigned as 1 to save memory. Except in the last step of multi-step training, where $\lambda_{detect}$ is set to 0.9 and $\lambda_{local}$ and $\lambda_{out}$ are set to 0.1, in the other steps, $\lambda_{detect}$, $\lambda_{basic}$, $\lambda_{BFA}$ and $\lambda_{local}$ are set to 1.

*C. Compare With SOTA Methods*

In this section, we compare the performance of our proposed low-resolution branch network of multi-step hybrid knowledge distillation with several state-of-the-art detection methods, including Hui Y, et al.(Faster RCNN), Hui Y, et al.(Cascade RCNN), FCOS, ATSS, DAB-DETR, and DINO. All these methods trained with high-resolution input and up sample the low-resolution input during inference to match the size of the high-resolution input. These detection methods encompass both one-stage and two-stage detection approaches as well as CNN and transformer-based detectors. Fig. 3 presents a subset of the obtained detection results. We can conclude that MHKD outperforms these state-of-the-art methods through the qualitative analysis of Fig. 3.

As shown in Table I, our baseline approach with single-step offline distillation performs outperforms Hui Y, et al., Hui Y, et al., FCOS, ATSS, DAP-DETR, and DINO on LN dataset. The student method of MHKD outperforms the baseline and the direct detection methods on the low-resolution images. The mean average precision $AP_{0.5:0.95}$ achieves an 23.1% improvement compared with the direct detection methods and 15.5% improvement compared with the baseline. While the MHKD method has higher complexity in terms of parameters, its inference time remains competitive with baseline methods and fully meets the clinical needs.

As shown in Table II, our baseline approach also outperforms Hui Y, et al., Hui Y, et al., FCOS, ATSS, DAP-DETR, and DINO on HUBMAP dataset. At the same time, the student method of MHKD achieves an improvement of 15.9% compared with the direct detection methods and 3.5% compared with baseline in the metric $AP_{0.5:0.95}$.

This shows that MHKD makes the student model effectively learn information from high-resolution branches, enabling it to extract more representative features from low-resolution inputs.

*D. Ablation Experiments*

In this section, we analyze the effect of each component in our method on LN dataset.

*1) Advantages of Basic Feature Adaptive Module:* We adapt a multi-step hybrid knowledge distillation training strategy, which conducts three-step distillation training (basic features, local features, semantic information) and four-step distillation training (basic features, adaptive features, local features, semantic information) for whether BFA is added or not. The Table III shows the influence of the Basic Feature Adaptive module on low resolution branch detection performance. The addition of the BFA module and the distillation of adaptive features improved the $AP_{0.5:0.95}$ by 6.8%, It is proved that the BFA module greatly improves the detection performance of low-resolution pathological images.

*2) Advantages of Hybrid Knowledge Distillation Strategy:* Online knowledge distillation and offline knowledge distillation are common knowledge distillation strategies. Here we compare our proposed hybrid knowledge distillation with online distillation and offline distillation on the basis of adding BFA and maintaining the multi-step distillation training strategy. As shown in Table IV, the detection performance of hybrid knowledge distillation is better than that of online distillation and offline distillation in all metrics. It can be seen that hybrid knowledge distillation plays a key role in our method.

*3) Advantages of Multi-Step Training Strategy:* Table V presents the detection results for different distilled locations. We perform knowledge distillation at the three locations of basic features, local features, and final output respectively, showing better results than no knowledge distillation, proving that knowledge distillation at all three locations is effective.

The training strategy is determined based on the distillation information partition. In single-step training strategy, we perform online distillation of multiple locations simultaneously. In double-step training strategy, in the first step, we perform offline distillation of global features (basic features and adaptive global features), and in the second step, we perform online distillation of semantic information. In four-step training strategy, that is, our proposed MHKD method. As shown in Table VI, the detection performance is significantly improved with the increase of the number of steps. This proves that even if the distillation is performed at all four locations, the training step by step is more effective to improve the detection performance of low-resolution pathological images.

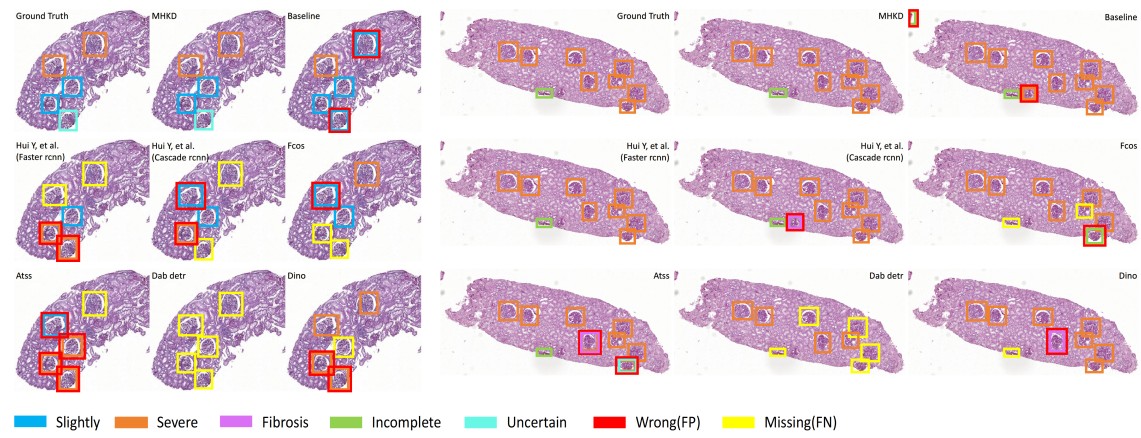

| | Slightly | | Severe | | Fibrosis | | Incomplete | | Uncertain | | Wrong(FP) | | Missing(FN) |

Fig. 3. Visualization of the results of the various methods. The red boxes are false positives and the yellow boxes are false negatives.

TABLE I
PERFORMANCE OF DIFFERENT DETECTION METHODS ON LN DATASET.

| Method | $AP_{0.5:0.95}$ | $AP_{0.5}$ | $AP_{0.75}$ | $AR_{0.5:0.95}$ | $AP_{Slight}$ | $AP_{Severe}$ | $AP_{Fibrosis}$ | $AP_{Incomplete}$ | $AP_{Uncertain}$ | $Params$ | $Time$ |
|---|---|---|---|---|---|---|---|---|---|---|---|
| FCOS [9] | 25.2% | 39.9% | 29.1% | 43.3% | 45.7% | 37.4% | 13.2% | 18.7% | 10.8% | 50.79M | 24.78s |
| ATSS [10] | 29.9% | 44.4% | 34.9% | 52.2% | 53.6% | 42.7% | 15.9% | 23.6% | 13.8% | 50.89M | 24.49s |
| DAB-DETR [19] | 29.1% | 42.8% | 34.7% | 61.3% | 60.5% | 48.7% | 7.1% | 22.2% | 6.9% | 62.59M | 2.22s |
| DINO [20] | 32.7% | 47.2% | 38.8% | 66.1% | 62.1% | 49% | 15.7% | 27.3% | 9.3% | 66.44M | 2.19s |
| Hui Y, et al. [6] | 22.3% | 31.5% | 27.5% | 34.2% | 39.8% | 31.8% | 8.8% | 18.3% | 12.8% | 60.14M | 22.84s |
| Hui Y, et al. [13] | 24.3% | 33.9% | 28.6% | 34.9% | 42.6% | 33.3% | 10.6% | 21.9% | 13.4% | 87.93M | 24.41s |
| Baseline(KD) | 40.3% | 60.6% | 47.7% | 55.7% | 63.6% | 47.1% | 37.1% | 37.6% | 16.0% | 47.33M | 6.27s |
| MHKD | 55.8% | 76.9% | 68.8% | 63.8% | 69.5% | 54.8% | 52.6% | 55.0% | 46.9% | 311.60M | 6.54s |

Here for comparison, all methods are trained on high-resolution pathology images, and the low-resolution input is up sampled during inference to align with the high-resolution input. Baseline(KD) is the baseline we propose for this novel task. In the baseline, the strategy is to perform a single-step offline knowledge distillation on basic feature, local feature, and output simultaneously. MHKD is the method we proposed for this novel task. In our method, the strategy is to perform hybrid knowledge distillation in four steps. The distillation locations of the four steps are the basic feature, the adaptive feature, the local feature, and the semantic information. Here, Time is the inference time.

TABLE II
PERFORMANCE OF DIFFERENT DETECTION METHODS ON HUBMAP DATASET.

| Method | $AP_{0.5:0.95}$ | $AP_{0.5}$ | $AP_{0.75}$ | $AR_{0.5:0.95}$ |
|---|---|---|---|---|
| FCOS [9] | 11.7% | 39.2% | 3.7% | 24.3% |
| ATSS [10] | 12.5% | 41.8% | 3.6% | 20.1% |
| DAB-DETR [19] | 19.4% | 37.4% | 17.1% | 26.6% |
| DINO [20] | 23.4% | 41.5% | 22.6% | 39.5% |
| Hui Y, et al. [6] | 11.4% | 24.5% | 8.6% | 13.1% |
| Hui Y, et al. [13] | 10.4% | 19.7% | 9.4% | 11.6% |
| Baseline(KD) | 35.8% | 76.9% | 25.8% | 44.9% |
| MHKD | 39.3% | 80% | 31.5% | 46% |

TABLE III
INFLUENCE OF THE BASIC FEATURE ADAPTIVE MODULE ON LOW-RESOLUTION BRANCH DETECTION PERFORMANCE.

| BFA | $AP_{0.5:0.95}$ | $AP_{0.5}$ | $AP_{0.75}$ | $AR_{0.5:0.95}$ |
|---|---|---|---|---|
| ✗ | 49.0% | 70.5% | 60.6% | 59.3% |
| ✓ | 55.8% | 76.9% | 68.8% | 63.8% |

Here ✗ indicates that the model utilizes the MHKD on basic feature, local feature, and semantic information (local feature and output) without the Basic Feature Adaptive module and adaptive global feature. Here, ✓ means adding the Basic Feature Adaptive module to ✗ and adding a step of distillation of adaptive features after basic feature distillation.

TABLE IV
INFLUENCE OF DIFFERENT KNOWLEDGE DISTILLATION STRATEGIES ON LOW-RESOLUTION BRANCH DETECTION PERFORMANCE.

| Strategy | $AP_{0.5:0.95}$ | $AP_{0.5}$ | $AP_{0.75}$ | $AR_{0.5:0.95}$ |
|---|---|---|---|---|
| Online | 49.5% | 67.7% | 59.3% | 60.6% |
| Offline | 54.3% | 74.6% | 65.8% | 63.7% |
| Hybrid | 55.8% | 76.9% | 68.8% | 63.8% |

TABLE V
INFLUENCE OF KNOWLEDGE DISTILLATION AT DIFFERENT LOCATIONS ON LOW-RESOLUTION BRANCH DETECTION PERFORMANCE.

| Global | Local | Output | $AP_{0.5:0.95}$ | $AP_{0.5}$ | $AP_{0.75}$ | $AR_{0.5:0.95}$ |
|---|---|---|---|---|---|---|
| ✗ | ✗ | ✗ | 37.00% | 54.70% | 43.50% | 55.30% |
| ✓ | ✗ | ✗ | 40.40% | 62.60% | 45.60% | 58.80% |
| ✗ | ✓ | ✗ | 41.40% | 63.60% | 49.20% | 54.50% |
| ✗ | ✗ | ✓ | 40.20% | 61.40% | 48.50% | 55.00% |

## V. CONCLUSION

In this paper, we propose a glomerulus detection task in low-resolution human kidney pathology images, aiming to alleviate the limitations of hospitals in underdeveloped areas without sufficient cost to acquire high-resolution WSIs for kidney pathology diagnosis.

To address this challenge, we introduce a novel MHKD approach for glomerulus detection in low-resolution patho-

TABLE VI
INFLUENCE OF MULTI-STEP TRAINING STRATEGY ON LOW-RESOLUTION
BRANCH DETECTION PERFORMANCE.

| Training Strategy | $AP_{0.5:0.95}$ | $AP_{0.5}$ | $AP_{0.75}$ | $AR_{0.5:0.95}$ |
|---|---|---|---|---|
| Single-step training strategy | 39.3% | 57.7% | 47.1% | 55.9% |
| Double-step training strategy | 45.9% | 68.0% | 54.5% | 57.3% |
| Four-step training strategy | 55.8% | 76.9% | 68.8% | 63.8% |

Single-step training strategy represents online distillation at multiple locations simultaneously. Double-step training strategy means that the training is divided into two parts: offline distillation and online distillation. Four-step training strategy is our proposed MHKD method.

logical images. The hybrid knowledge distillation strategy combines offline and online distillation strategy, ensuring the feature extraction ability of the front-end, but also reduces the gap between the student model and the teacher model in the back-end. The multi-step training strategy facilitates the gradual transfer of high-resolution image information to the student model, from global features to semantic details. The MHKD effectively enhances the model's detection performance on low-resolution images compared to various direct detection methods, methods designed for high-resolution pathology images and simple distillation techniques. Future research will explore the integration of knowledge distillation into transformer-based object detection methods to further improve the accuracy of glomerulus detection in low-resolution pathological images and its clinical diagnostic potential.

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
