# OpenReview forum: "MHKD: Multi-step Hybrid Knowledge Distillation for Low-resolution Whole Slide Images Glomerulus Detection"
_IEEE.org/EMBS/BHI/2024/Conference — IEEE BHI'24_

### Official Review · Reviewer_FEQy · 2024-08-12
**Multi-step Hybrid Knowledge Distillation for Low-resolution Whole Slide Images Glomerulus Detection**

**Overall Rating:** 7
**Confidence:** 5

**Other Quality Metrics:**

(a) Clarity of writing - Great
 (b) Clinical Significance - Great
(c) Methodological Novelty - Great
(d) Experiments and Results - Great

**Questions For The Authors:**

1. Should we include the comparison of model complexity between models in this study?

**Strengths:**

This work introduces a hybrid knowledge distillation method that effectively transfers both global features and semantic information from high-resolution pathological images to a student model through a multi-step training strategy, integrating offline and online distillation techniques.

**Summary Of The Paper:**

This paper proposes a novel approach for glomerulus detection based on hybrid knowledge distillation from low-resolution images.

**Weaknesses:**

The only thing the reviewer wants to see is how the complexity of the model (training times, inference times) compared to relevant studies referenced by the authors.

Based on that, we will verify whether the model can be employed to address computational resource constraints in a hospital environment.

---

### Official Review · Reviewer_QGAv · 2024-08-16
**In-depth, engaging, and clinically relevant - MHKD: Low-Res Glomerulus Detection**

**Overall Rating:** 8
**Confidence:** 4

**Other Quality Metrics:**

(a) Clarity of writing - Great
(b) Clinical Significance - Great
(c) Methodological Novelty - Excellent
(d) Experiments and Results - Great

**Questions For The Authors:**

Overall the paper presents a very promising research in this area of KD in histopathology. The presentation is in-depth and engages the readers very well. However, it is curious why certain other works where they have used KD in histopathology have not been compared against, especially [6] that is in the reference, but not cited elsewhere in the paper. It would be meaningful if this detail is also provided.

There are a few other questions as well:

1. How does the inference time of MHKD compare to the baseline methods? Is there a significant computational cost to the multi-step approach?

2. How sensitive is the performance to the downsampling factor between high and low resolution images? Is there an optimal ratio?

**Strengths:**

This paper addresses an important clinical need for diagnosis in resource-constrained settings, especially using low resolution WSI.
The major strengths of the paper include:
1. Novel combination of offline and online knowledge distillation techniques
2. Systematic ablation studies to validate each component of the method
3. Comprehensive comparison against multiple state-of-the-art object detection approaches
4. Evaluation on both public and private datasets

**Summary Of The Paper:**

This paper proposes a novel method called Multi-step Hybrid Knowledge Distillation (MHKD) for glomerulus detection in low-resolution kidney pathology images. The key contributions are:

1. Introducing the task of glomerulus detection on low-resolution kidney pathology images
2. Proposing a hybrid knowledge distillation approach combining offline and online distillation
3. Developing a multi-step training strategy to transfer knowledge from high to low resolution models
4. Demonstrating improved performance over state-of-the-art methods on two datasets

They have demonstrated SOTA performance in this task.

**Weaknesses:**

It would be helpful to the readers if the authors include the following additional points:

1. Inference speed
2. Lack of comparison to other knowledge distillation approaches for medical imaging - especially reference [6] that the authors have included in the references but have not used in the comparison.
3. A comment on the relatively small datasets used for evaluation - and what are the expectation in terms of generalizability of the model on larger datasets

---

### Official Review · Reviewer_vR3h · 2024-08-19
**Enhancing Glomerulus Detection in Low-resolution WSIs with Multi-step Hybrid Knowledge Distillation**

**Overall Rating:** 7
**Confidence:** 5

**Other Quality Metrics:**

(a) Clarity of Writing: Good
(b) Clinical Significance: Excellent
(c) Methodological Novelty: Excellent
(d) Experiments and Results: Excellent

**Questions For The Authors:**

1. The MHKD framework shows promise in transferring high-resolution image labels to low-resolution images. How do you anticipate the MHKD method would perform on different types of histopathological images, such as those from other organs or diseases? Have you considered testing it on additional datasets?
2. The Basic Feature Adaptive (BFA) module appears to play a significant role in improving detection performance. Can you elaborate on the design choices behind this module and whether alternative designs were considered?
3. Given the complexity of the method, how do you envision integrating it into clinical workflows? Are there plans to develop a more streamlined or user-friendly version for clinical use?
4. Could the authors provide more details about the upsampling process mentioned in the method?

**Strengths:**

1. The authors effectively highlight the limitations of current methods, providing clear motivation for the study.
2.  The MHKD model employs a multi-step hybrid knowledge distillation method that effectively addresses the challenges of detecting glomeruli in low-resolution pathology images..
3. Rigorous evaluation on both private and public datasets demonstrates the method’s robustness and reliability. The authors also explore the model’s performance with and without specific components, highlighting its optimized design.
4. Experimental results indicate significant improvements in detection accuracy, with detailed comparisons to state-of-the-art methods, making it a strong candidate for clinical use.

**Summary Of The Paper:**

The manuscript introduces MHKD, the first approach to address the challenge of glomerulus detection in low-resolution whole slide images for kidney pathology. This method reduces reliance on high-resolution digital pathology images, which are resource-intensive in terms of equipment, storage, and computational power. MHKD combines offline and online knowledge distillation techniques, transferring global and semantic features from a high-resolution teacher model to a low-resolution student model. By applying a multi-step training strategy, the model ensures that information from high-resolution images is gradually and effectively transferred to the student model, further enhancing the distillation effect. Results from the two independent datasets demonstrate that MHKD significantly improves detection performance compared to current state-of-the-art methods.

**Weaknesses:**

1. The overall writing style could be improved to enhance clarity and accessibility for a broader audience.
2. The HuBMAP dataset includes only 20 kidney images, suggesting that a larger dataset would be beneficial for more robust validation.
3. While the authors mention the computational cost of analyzing kidney images, the manuscript lacks a benchmark comparison of the computational efficiency of their model.

---

### Decision · Program_Chairs · 2024-09-23

Accept